# Effects of Progressive Drought Stress on the Growth, Ornamental Values, and Physiological Properties of *Begonia semperflorens*

Zhimin Zhao [1], Airong Liu [2,*], Yuanbing Zhang [3], Xiaodong Yang [4], Shuyue Yang [1] and Kunkun Zhao [3,*]

[1]   College of Agriculture, Anhui Science and Technology University, Donghua Road No. 9,
      Chuzhou 233100, China; 19556958506@163.com (Z.Z.); 19565985302@163.com (S.Y.)
[2]   College of Life and Health Sciences, Anhui Science and Technology University, Donghua Road No. 9,
      Chuzhou 233100, China
[3]   College of Architecture, Anhui Science and Technology University, Donghua Road No. 9,
      Chuzhou 233100, China; zyb2246@163.com
[4]   College of Environmental Ecology, Jiangsu Open University, Jiangdong North Road No. 399,
      Nanjing 210036, China; yangxd@jsou.edu.cn
*    Correspondence: liuar@ahstu.edu.cn (A.L.); kunkunzhao2021@163.com (K.Z.);
     Tel.: +86-(0552)-3197206 (A.L. & K.Z.)

**Abstract:** Water is one of the most important elements affecting the growth of ornamental plants. To investigate the effects of drought stress on the growth, ornamental values, and physiological properties of *Begonia semperflorens*, watering treatments with 250 mL (control check, CK), 200 mL (extremely light drought, ELD), 150 mL (light drought, LD), 100 mL (moderate drought, MD), 50 mL (severe drought, SD), and 25 mL (extremely severe drought, ESD) on the *B. semperflorens* variety "Chao Ao" were performed in this study. As a result, compared to the control (CK), the number of flowers, leaves, and branches, leaf size, plant height, crown diameter, as well as water content, transpiration rate, net photosynthetic rate, stomatal conductance, intercellular $CO_2$ concentration, and chlorophyll content in leaves decreased, followed by an increased amount of drought stress. The contents of the osmotic adjustment substances, such as soluble sugar, soluble protein, proline, and betaine, were increased under drought stress. Indicators related to antioxidant activities, such as SOD activity, increased and then decreased. The POD activity, CAT activity, MDA content, and plasma membrane permeability of *B. semperflorens* were higher under increased drought stress than in the control condition. The APX activity decreased and then increased under drought stress. In conclusion, *B. semperflorens* responds to drought stress by increasing osmotic adjustment substances and antioxidant activities and reducing the water loss, growth potential, and photosynthetic rate. The correlation analysis showed that, except for APX, the drought resistance coefficients of 23 other indexes were correlated in different degrees. Therefore, this study suggests that *B. semperflorens* has a strong drought resistance ability, retaining high ornamental values in conditions of moderate drought stress, and can still survive under extremely high drought stress.

**Keywords:** *Begonia semperflorens*; drought stress; growth; ornamental values; physiological properties





## 1. Introduction

Global climate change increases the frequency and intensity of drought, which is one of the important environmental issues all around the world [1,2] and is harmful to the growth of ornamental plants. Because water is important for the growth of ornamental plants, an important research focus has been on water-saving management during the cultivation of ornamental plants [3]. Therefore, to save costs and improve the ecological benefits, it is practically meaningful to choose herb ornamental plants with better drought resistance and high ornamental values for arid areas.

A lack of water can result in leaf abscission and even the death of a whole plant, so plants normally respond to drought for survival by changing morphologies, such as

lowering their plant height, leaf size, and ground diameter [4,5]. Research has shown that during drought stress, plants will accumulate more reactive oxygen and have higher peroxidization of the cytomembrane. Therefore, the photosynthesis process will be affected; there will be a lack of energy supply for metabolism, resulting in the stunted growth of plants [6–9]. SOD and POD activities and chlorophyll content are important indicators for evaluating the drought resistance abilities of interspecific hybrids of *Elymus* L. [10]. Drought affects the growth of winter wheat by affecting the balance of carbon and nitrogen metabolism [11]. For now, flowers with high ornamental values and strong resistance are essential for urban landscaping [12].

*Begonia semperflorens*, belonging to Begoniaceae, is an evergreen perennial herb and is the most common species in *Begonia*. It is a beautiful plant type with a long flowering period and colorful flowers. In addition, it blooms all year round and has many flowers. These characteristics show the high commercial value of *B. semperflorens*. Therefore, *B. semperflorens* has been widely used in landscaping and is popular in flower applications. Until now, the research on *B. semperflorens* has been about cultivation techniques [13,14], the response to light intensity [15,16], and the response to low temperatures [17]. An *MYB* gene, *BsMYB*62, in *B. semperflorens* was found to be involved in responding to drought stress via transgenic *Arabidopsis thaliana* [18]. However, the physiological mechanism of *B. semperflorens* responding to drought stress remains unclear. Therefore, in this study, the changes in growth, ornamental values, and physiological properties in the *B. semperflorens* variety "Chao Ao" under different degrees of drought stress were analyzed. The results will provide a theoretical basis for promoting drought resistance and provide a reference for scientific and rational irrigation in arid regions or arid seasons.

## 2. Materials and Methods

### 2.1. Plant Materials and Growth Conditions

This study was performed in the experimental base of Anhui Science and Technology University (112°261770′, 39°741886′). Three seedlings (4~5 leaves, 8.4 cm in plant height) of the *B. semperflorens* variety "Chao Ao" were cultivated in each flowerpot (11 cm in height, 21 cm in diameter). Each treatment contained at least 10 flowerpot plants. The substrate was silver sand, the pH was 6.8–7.4, and the porosity was approximately 40%. The plants were grown in a greenhouse under natural conditions. The testing occurred from 15 March 2023 to 10 July 2023. Hoagland's nutrient solution (Hoagland: water = 1:20, volume ratio) was used for the irrigation every three days [19], and the irrigation occurred between 17:00 and 18:00 pm. These plants were cultivated under consistent conditions.

### 2.2. Drought Treatment

After 60 days of cultivation, the plants with the same growth conditions were used for the drought treatments. There were six groups for treatments, including the control (CK, normal water quantity, 250 mL), extremely light drought condition (ELD, 80% of normal water quantity, 200 mL), light drought condition (LD, 60% of normal water quantity, 150 mL), moderate drought condition (MD, 40% of normal water quantity, 100 mL), severe drought condition (SD, 20% of normal water quantity, 50 mL), and extremely severe drought condition (ESD, 10% of normal water quantity, 25 mL). After 25 days, the morphological and physiological indexes of the plants for each treatment were measured.

### 2.3. Determination of the Photosynthetic Indexes

The chlorophyll (chl) content was measured via an acetone extraction method [20]. The TPS-2 portable photosynthesis system (PP SYSTEMS, Amesbury, MA, USA) was used to measure the net photosynthetic rate (Pn), transpiration rate (Tr), stomatal conductance (Gs), and intercellular $CO_2$ concentration of the leaves at 9:00~11:00 am.

### 2.4. Determination of the Osmotic Adjustment Substances and Indicators Related to Antioxidant Ability

The leaf water content (LWC) and stem water content (SWC) were measured via an oven-drying method. The soluble protein (SP) was measured via Coomassie brilliant blue staining. The betaine content was measured using a colorimetric method [20]. The soluble sugar (SS) content was measured using an anthrone method, the proline (Pro) content was measured using the ninhydrin method, the malondialdehyde (MDA) content was measured via the thiobarbituric acid method, and the membrane permeability was measured using a conductivity meter [19]. The superoxide dismutase (SOD) activity was determined via the nitroblue tetrazolium (NBT) method, and the inhibition of NBT photochemical reduction by 50% was expressed as one SOD activity unit (U) [16]. The peroxidase (POD) activity was determined via the guaiacol method, and the catalase (CAT) activity and ascorbate peroxidase (APX) activity were determined via ultra-violet absorption spectrometry [20]. Three biological duplications were performed in each experiment, and three flowerpots of plants were used in each duplication. The samples were used for all the analyses. In addition, each duplication was 0.5 g, but 10 g was used for testing the LWC and SWC.

### 2.5. Statistical Analysis

To calculate the statistics and create the graph, we used Excel 2019. The variance analysis of data was performed by using Duncan's test at $p < 0.05$ in SPSS 27.0. The R programming language was used for the z-score normalization of 24 indexes, and then heatmap software and corrplot software were used to draw the heat map and Pearson correlation heat map, respectively [21].

## 3. Results

### 3.1. Effect of Gradient Drought Stress on the Growth of B. semperflorens

Figure 1 shows that the ornamental values of *B. semperflorens* decreased during the increased levels of drought stress. In treatments of SD and ESD, the plants almost lost their ornamental values, with few flowers, a pale floral color, and curly and thinner leaves. The statistical analysis of the morphological indexes (Table 1) shows that compared to the CK, the flower number (FN), leaf number (LN), branch number (BN), leaf length (LL), leaf width (LW), plant height (PH), and crown diameter (CD) of the plants in different drought treatment were decreased. The results show that *B. semperflorens* still had a 100% survival rate with the ESD treatment, which only provided 25 mL of water. It suggests that *B. semperflorens* has strong drought resistance.

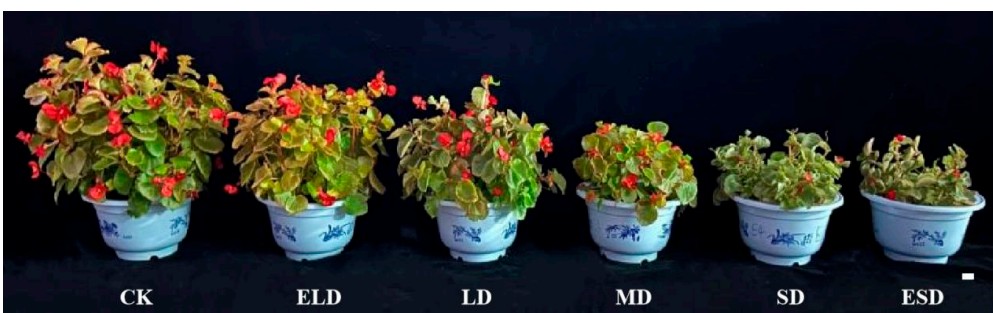

**Figure 1.** Phenotypes of *B. semperflorens* under drought stress; control check, extremely light drought, light drought, moderate drought, severe drought, and extremely severe drought conditions represent CK, ELD, LD, MD, SD, and ESD, respectively. Bar = 2 cm.

**Table 1.** Analysis of the morphological indexes of *B. semperflorens* under a gradient of drought stress.

| Treatment | FN | LN | BN | LW (cm) | LL (cm) | PH (cm) | CD (cm) |
|---|---|---|---|---|---|---|---|
| CK | 73 ± 2.05 a | 197 ± 0.82 a | 29 ± 0.47 a | 5.4 ± 0.12 a | 5.5 ± 0.04 a | 21.4 ± 0.52 a | 35.3 ± 0.21 a |
| ELD | 52 ± 1.63 b | 179 ± 2.16 b | 26 ± 0.47 b | 4.8 ± 0.08 b | 5.2 ± 0.12 b | 18.1 ± 0.16 b | 34.8 ± 0.58 b |
| LD | 31 ± 0.47 c | 159 ± 0.81 c | 24 ± 0.81 c | 4.4 ± 0.04 c | 4.3 ± 0.08 c | 15.9 ± 0.43 c | 32.6 ± 0.21 c |
| MD | 23 ± 2.05 d | 142 ± 0.81 d | 21 ± 0.81 d | 3.9 ± 0.08 d | 4.1 ± 0.21 d | 13.6 ± 0.29 d | 27.1 ± 0.33 d |
| SD | 11 ± 0.82 e | 105 ± 0.81 e | 17 ± 0.47 e | 3.4 ± 0.08 e | 3.3 ± 0.08 e | 11.2 ± 0.21 e | 23.9 ± 0.33 e |
| ESD | 5 ± 0.82 f | 88 ± 2.05 f | 13 ± 0.94 f | 2.7 ± 0.08 f | 2.9 ± 0.21 f | 9.5 ± 0.43 f | 20.6 ± 0.90 f |

a–f: mean ± standard error, with the different lowercase letters in the same column, indicating a statistically significant difference at a $p < 0.05$ level.

### 3.2. Effect of Gradient Drought Stress on the Water Content and Transpiration Rates in the Leaves and Stems of B. semperflorens

During the increased levels of drought stress, the plants showed decreased water content in the leaves and stems (Figure 2A). The water contents in leaves were 95.72%, 95.22%, 93.41%, 92.93%, 92.61%, and 89.10%, respectively. The water contents in the stems were 94.13%, 93.05%, 91.38%, 90.40%, 89.01%, and 86.35%, respectively. The water content in the leaves and stems under ESD was 6.92% and 8.27% less than the CK, respectively. The transpiration rates in the leaves were 4.67, 4.17, 3.67, 1.48, 0.92, and 0.38 mmol·m$^{-2}$·s$^{-1}$, showing similar trends of water content (Figure 2B). The transpiration rates in the leaves under ELD and ESD were 11% and 92% lower than the CK, respectively.

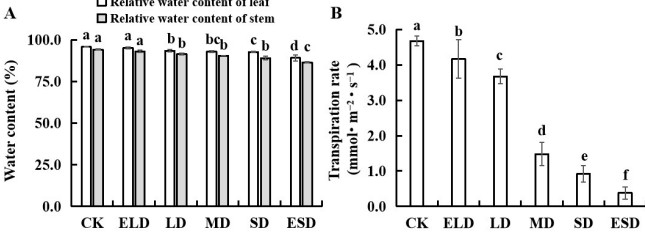

**Figure 2.** Relative water content in the leaves and stems (**A**) and transpiration rates in the leaves (**B**) of *B. semperflorens* under drought stress. The values are the mean ± SE (*n* = 3). Significant differences were analyzed using Duncan's multiple range test; the bars with different letters are significantly different from each other ($p < 0.05$).

### 3.3. Effect of Gradient Drought Stress on the Photosynthetic Indexes of B. semperflorens Leaves

The increased levels of drought stress induced changes in the chlorophyll content of the leaves with a decreasing tendency, which were 0.37 mg/g, 0.35 mg/g, 0.31 mg/g, 0.22 mg/g, 0.20 mg/g, and 0.17 mg/g, respectively (Figure 3A). The chlorophyll content in the leaves under ESD was 52.92% less than the CK. Meanwhile, the net photosynthetic rate, intercellular $CO_2$ concentration, and stomatal conductance also showed a decreasing tendency (Figure 3B–D). The net photosynthetic rates in the leaves of the other five treatments were 0.21%, 0.33%, 0.48%, 0.62%, and 0.79% lower than the CK (Figure 3B). The intercellular $CO_2$ concentrations in the leaves of the other five treatments were 21.75%, 42.13%, 46.63%, 49.50%, and 63.94% less than the CK, whereas there was no significant difference between the intercellular $CO_2$ concentrations in the leaves under MD and SD (Figure 3C). The stomatal conductance under five treatments of drought stress declined by 14.77%, 38.94%, 70.84%, 89.30%, and 96.98%, compared to the CK (Figure 3D).

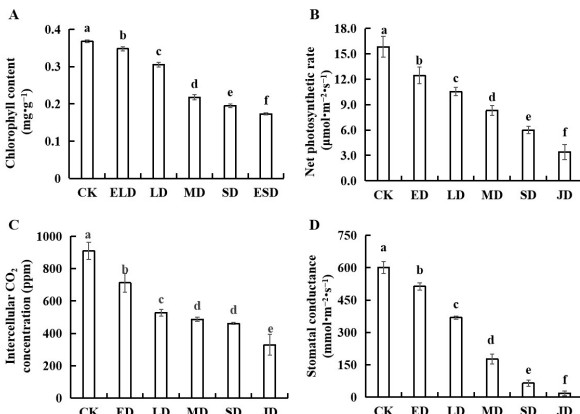

**Figure 3.** The chlorophyll content (**A**), photosynthetic rate (**B**), intercellular $CO_2$ concentration (**C**), and stomatal conductance (**D**) in the *B. semperflorens* leaves under drought stress. Values are the mean ± SE (*n* = 3). Significant differences were analyzed using Duncan's multiple range test; the bars with different letters are significantly different from each other (*p* < 0.05).

### 3.4. Effect of Gradient Drought Stress on the Osmotic Adjustment Substances in *B. semperflorens* Leaves

The osmotic adjustment substances in the *B. semperflorens* leaves under drought stress showed an increasing tendency. Under the ELD condition, the contents of soluble protein and betaine were higher than those of the CK (Figure 4A,B), whereas the contents of proline and soluble sugar showed no differences compared to the CK (Figure 4C,D). It indicates that soluble protein and betaine may be more sensitive to drought stress. From ELD to ESD, four osmotic adjustment substances increased (Figure 4).

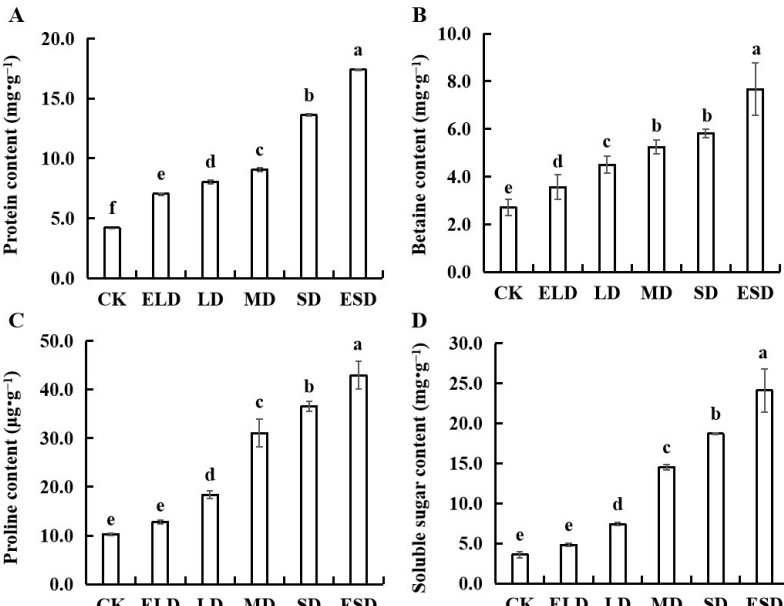

**Figure 4.** The osmotic adjustment substance protein content (**A**), betaine content (**B**), proline content (**C**), and soluble sugar (**D**) in *B. semperflorens* leaves under drought stress. Values are the mean ± SE (*n* = 3). Significant differences were analyzed using Duncan's multiple range test; the bars with different letters are significantly different from each other (*p* < 0.05).

Further analyses showed that compared to those of the CK, the soluble protein content under treatments from ELD to ESD increased by 67.46%, 91.45%, 115.20%, 223.04%, and 313.30%; the betaine content increased by 31.79%, 66.29%, 93.89%, 115.11%, and 183.84%; the proline content increased by 24.57%, 79.53%, 202.92%, 256.31%, and 318.76%; and the soluble sugar content increased by 33.33%, 105.56%, 302.78%, 419.44%, and 569.44%. When

comparing between two adjacent treatments, the soluble protein content between MD and SD showed the largest increase, which was 50.11%; the betaine content between SD and ESD showed the largest increase, which was 31.96%; the proline content between LD and MD showed the largest increase, which was 32.27%; and the soluble sugar content between LD and MD showed the largest increase, which was 95.95%.

### 3.5. Effect of Gradient Drought Stress on the Antioxidant Activities in B. semperflorens Leaves

The membrane permeability, MDA content, CAT activity, and SOD activity in *B. semperflorens* leaves under drought stress almost showed an increasing tendency (Figure 5A–C,E). Compared to the CK, the membrane permeabilities under treatments from ELD to ESD increased by 40.48%, 65.48%, 76.19%, 110.71%, and 115.48% (Figure 5A); the MDA content increased by 4.14%, 11.49%, 9.50%, 13.59%, and 11.38% (Figure 5B); and the CAT activities increased by 1.53–6.26 times (Figure 5C). The APX activities under ELD and LD were 12.50% and 25.00% lower than the CK, whereas the APX activities under MD, SD, and ESD increased by 12.50–62.50% (Figure 5D).

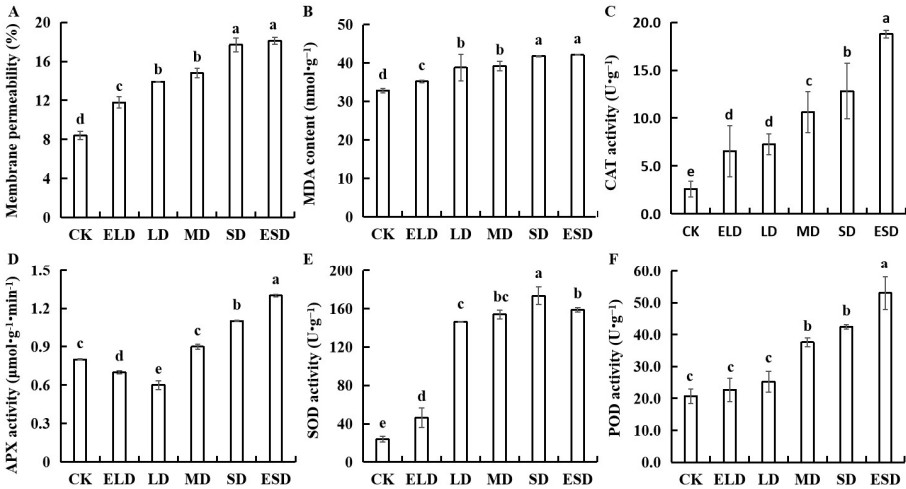

**Figure 5.** Effect of drought stress on the membrane permeability (**A**), MDA content (**B**), CAT activity (**C**), APX (**D**), SOD activity (**E**), and POD activity (**F**) in *B. semperflorens* leaves. Values are the mean ± SE (*n* = 3). Significant differences were analyzed using Duncan's multiple range test; the bars with different letters are significantly different from each other (*p* < 0.05).

During increasing levels of drought stress, the SOD activities increased by 94.14–625.52%. In comparison between two adjacent treatments, the SOD activities under LD were 215.30% higher than those under ELD, showing the largest increase; the SOD activities under ESD were 9.77% lower than those under SD, but 499.56% higher than those under the CK (Figure 5E). It suggests that the SOD activities responded to drought stress quickly but showed less response under ESD. Compared to the CK, the POD activities from ELD to ESD increased by 9.66–156.04%. In comparison between two adjacent treatments, the POD activities between LD and MD showed the largest increase, which was 49.21%, whereas there were no significant differences between LD, ELD, and CK (Figure 5F). These results suggest that the POD activities responded to drought stress slowly.

### 3.6. Correlation Analysis between the Indexes of Drought Resistance of B. semperflorens

Figure 6 shows that, except for APX, there are relationships between the 23 indexes in different degrees. Among these indexes, there were 31 pairs of indexes, such as FN and LN, FN and BN, and chl and Ci, showing significantly positive correlations (*p* < 0.05); 35 pairs of indexes, such as Gs and SP, Gs and CAT, and BC and SOD showed significantly negative correlations (*p* < 0.05); 67 pairs of indexes, such as FN and Pn, FN and Gs, and LN and chl showed significantly positive correlations (*p* < 0.05); 68 pairs of indexes, such as chl and BC, chl and MDA, and Ci and BC showed extremely significantly negative correlations

($p < 0.01$); 24 pairs of indexes, such as FN and Ci, LN and Pn, and BC and CAT showed extremely significantly positive correlations ($p < 0.01$); and 22 pairs of indexes, such as LN and SPH, LN and POD, and Pn and CAT showed extremely significantly negative correlations ($p < 0.01$).

Figure 7 shows that according to the sensibility of the response to drought stress, 24 indexes could be divided into three groups: APX has strong sensitivity; SOD, MDA, MP, SP, BC, CAT, Pro, POD, and SS have relatively strong sensitivity; and LWC, FN, Ci, LL, LN, SWC, Pn, LW, BN, Gs, CD, PH, TR, and chl have less sensitivity.

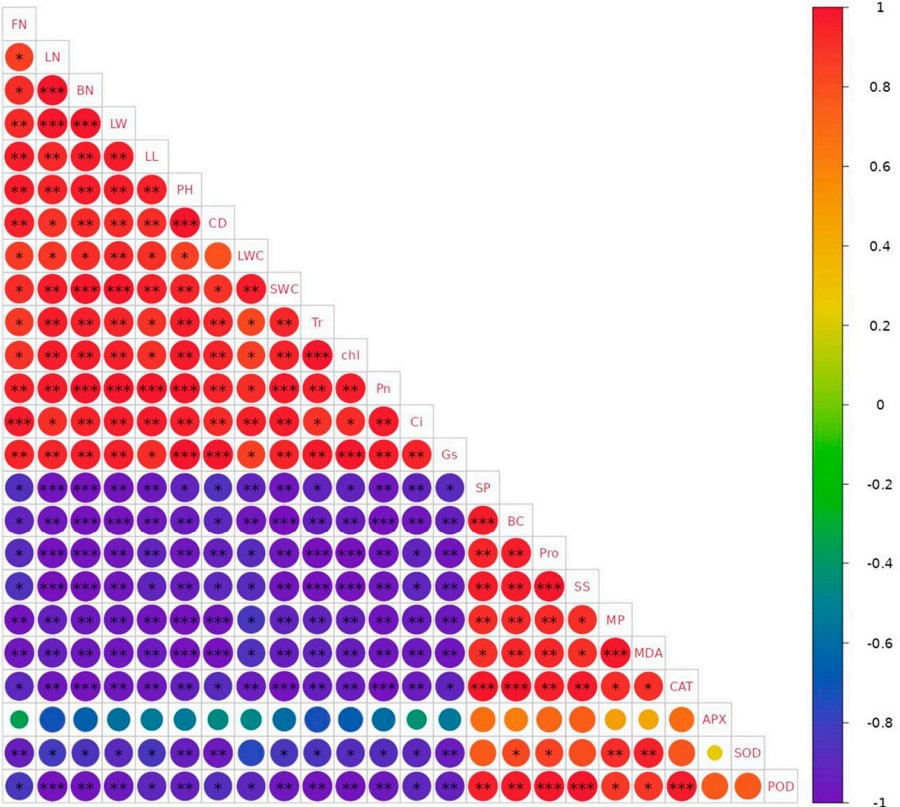

**Figure 6.** Correlation analysis between 24 indexes of *B. semperflorens*. *, **, and *** indicate significant correlation at levels of $p = 0.05$, $p = 0.01$, and $p = 0.001$, respectively. Control check, extremely light drought, light drought, moderate drought, moderate drought, and extremely severe drought represent CK, ELD, LD, MD, SD, and ESD, respectively.

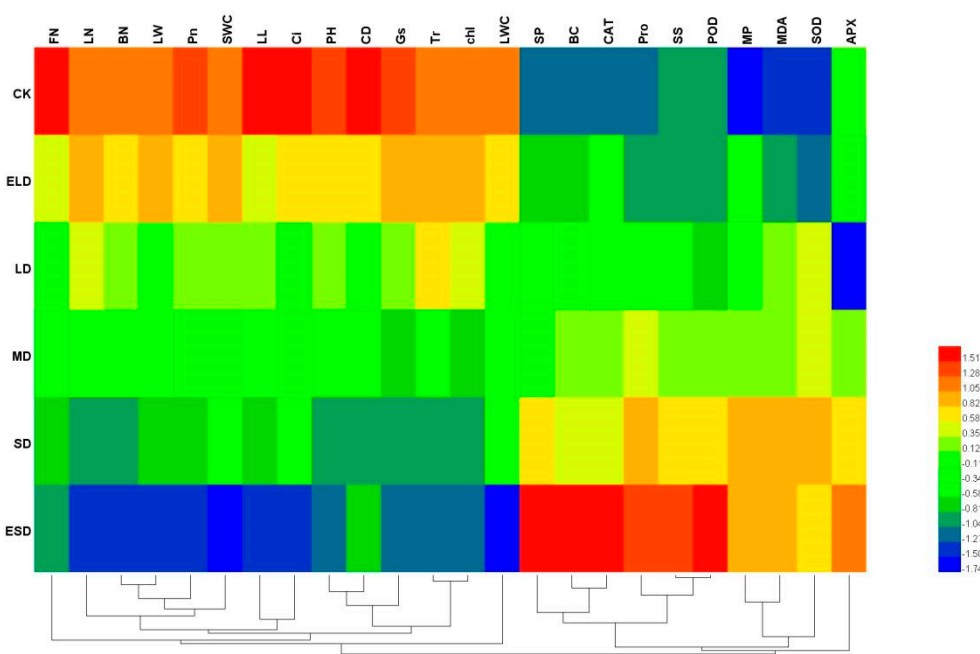

**Figure 7.** Heat map analysis of 24 indexes of *B. semperflorens.* Control check, extremely light drought, light drought, moderate drought, moderate drought, and extremely severe drought represent CK, ELD, LD, MD, SD, and ESD, respectively.

## 4. Discussion

Under drought conditions, the ability of plants to adapt to the changes in the soil moisture content by slowing growth or changing morphological characteristics is an important survival strategy [5]. In the present study, during the increased levels of drought stress, the plant height, crown diameter, leaf length, leaf width, leaf number, and flower number of *B. semperflorens* declined significantly (Figure 7), and the leaves became curly (Figure 1). Plants such as *Chloris virgata* adapting to drought stress by changing morphological characteristics have also been reported previously [22]. *Rhododendron delavayi* adapts to drought stress by reducing plant height [5]. *Medicago sativa* reduces its growth rate in response to drought stress [23]. Plants could reduce transpiration by reducing leaf area [24,25]. The curly leaves of *B. semperflorens* under drought stress were probably for saving water content in plants by preventing water consumption and reducing the temperature of the leaf surface. Similar research was reported in other plants [25,26]. Drought stress also reduced the water content in the leaves and stems and the transpiration rate in the leaves of *B. semperflorens* (Figure 2). Similar research was reported in plants such as *Chrysanthemum morifolium*, *Rhododendron delavayi*, and *Helleborus orientlis* [5,6,25]. The distribution of water in plants is normally based on the metabolic capability of the plant tissues. Therefore, for the survival of plants, the plant tissues with strong metabolic capability may contain more water [27].

Chlorophylls are basic substances for photosynthesis in leaves, and the chlorophyll concentrations affect the photosynthetic capacity of plants. In the present study, we found that the chlorophyll content declined when the levels of drought stress increased, which was consistent with the repression of the growth of *B. semperflorens*. Stomatal conductance affects the transpiration and photosynthesis of plants [28]. Drought stress reduced the net photosynthetic rate, transpiration rate, and stomatal conductance in the leaves of *Malus halliana* [29]. The reduced stomatal conductance in the leaves of *B. semperflorens* in our study may be a strategy of the plants for responding to drought stress [30]. Under drought stress, the net photosynthetic rate of *Helleborus orientlis* and *Camellia reticulate* also declined in previous studies [25,31].

Under drought stress, osmotic adjustment substances will accumulate to maintain the cell osmotic pressure of plants and to ensure the orderly conduct of a series of physiological

and biochemical reactions in plant cells [32,33]. Osmotic adjustment substances protect macromolecular substances. For example, soluble proteins can maintain the integrality of proteins [34]. Osmotic adjustment substances also provide carbon and nitrogen sources for plants [35]. Our present study found that the contents of soluble protein, soluble sugar, proline, and betaine significantly increased with increasing levels of drought stress (Figure 7). Increased osmotic adjustment substances can maintain the water balance in cells by promoting the water-retaining property of cells [36]. In the present study, for adapting to drought stress, increased osmotic adjustment substances in *B. semperflorens* could slow down membrane lipid peroxidation, maintain permselectivity, and promote water-retaining properties [22,25,37]. It suggests that *B. semperflorens* has a strong drought-resistance ability. The leaves of Sect. *Aigeiros* clones of *Populus*, *Populus wutunensis*, *Populus euramericana*, *Populus balleana*, and *Zea mays* also respond to drought stress by accumulating osmotic adjustment substances [38–40].

Normally, the cytomembrane permselectivity of plants is relatively stable [41]. However, drought stress may destroy the cell membrane system and induce membrane lipid peroxidation, damaged cytomembranes, and increased permeability [25,38]. As shown in Figure 5, cell membrane permeability and membrane lipid peroxidation products like MDA in *B. semperflorens* increased following the increased levels of drought stress. This phenomenon is similar to research about *Hemerocallis fulva*, *Helleborus orientlis*, and *Cyclocarya paliurus* [25,42,43]. Therefore, the increase in the cell membrane permeability indicates damage to plants induced by drought stress. The antioxidant defense system is important for plants to maintain and protect the metabolic balance of reactive oxygen [25]. Superfluous superoxide anion radical induced by abiotic stress can be catalyzed into $H_2O_2$ by SOD [44], and then $H_2O_2$ can be oxidized by POD, which is the electron acceptor of $H_2O_2$ [45]. APX can eliminate $H_2O_2$ [46]. We found that the activities of SOD and POD in *B. semperflorens* leaves were activated by drought stress and increased following the levels of drought stress, while the APX activity increased after MD treatment. It suggests that the sensitivities of SOD, POD, and APX activities in *B. semperflorens* to drought stress are different, which were also reported in *Hydrangea macrophylla* and *Helleborus orientlis* [25,47]. However, the SOD activity in *B. semperflorens* leaves declined under ESD treatment (Figure 5). LD treatment probably promoted the expression of SOD genes, then induced the increase in SOD activity and the production of $H_2O_2$. However, the increased drought stress destroyed the metabolic balance, which down-regulated the expression of the SOD genes and SOD activity. On the other hand, the activities of APX, CAT, and POD, which are responsible for eliminating $H_2O_2$, retained an increasing tendency, which is consistent with research results on *Helleborus orientlis* [25].

There are similar indexes affecting the ornamental values of *B. semperflorens*. The correlation analysis showed that FN, LN, BN, LW, LL, PH, and CD were positively correlating with photosynthetic indexes like chl, Tr, and Gs but negatively correlating with osmotic adjustment substances (SP, BC, Pro, and SS) and antioxidase (CAT, SOD, and POD) activities. Similar results were found in research on *Miscanthus sinensis* "Variegatus" and *Elymus sibiricus* [10,48]. Interestingly, salicylic acid, titanium dioxide nanoparticles, and sodium hydrogen sulfide could improve the inhibition of water deficiency on the ornamental quality of periwinkle [49].

In this study, *B. semperflorens* most sensitively responded to drought stress by strengthening the antioxidant ability to avoid oxidative damage and accumulating osmotic adjustment substances to increase the water-retention capacity [21]. By reducing the flower number, growth rate, and photosynthetic capacity, *B. semperflorens* reduced its energy waste, water requirement, and water desorption and slowed down the reduction of its autotrophic capacity. It is worth mentioning that among these morphological indexes, reduced flower number was the most sensitive to drought stress. Therefore, it suggests that compared to vegetative growth, reproductive growth is more sensitive to drought stress. The above results may provide physiological and biological direction for the cultivation of *B. semperflorens* in arid environments.

## 5. Conclusions

Our results showed that the levels of sensitivity of the *B. semperflorens* indexes to drought stress are different, which, from strong to weak, are as follows: increased antioxidant ability and accumulation of osmotic adjustment substances, reduced flower number, reduced growth, and photosynthetic capacity. Under moderate drought (40% of normal water quantity, 100 mL), *B. semperflorens* still had relatively high ornamental values. Under extremely severe drought (10% of normal water quantity, 25 mL), *B. semperflorens* still survived. Therefore, *B. semperflorens* has strong drought tolerance and is suitable for use in ecological and landscape construction in arid regions or arid seasons.

**Author Contributions:** A.L., K.Z., Y.Z., Z.Z., S.Y. and X.Y. performed the experiments. A.L. and Y.Z. conceptualized and supervised the research. K.Z. and A.L. participated in the writing of the manuscript. All authors have read and agreed to the published version of the manuscript.

**Funding:** This research was funded by the Key Projects of Natural Science Research in the colleges and universities of Anhui (2022AH051621, 2022AH051622), the Science and Technology Plan Project of Anhui Housing Construction Department (2022-YF038), Key Research Project of Chuzhou (2022ZN015), introduction of talent projects of Anhui Science and Technology University (JZYJ202201), and the Natural Science Foundation of Zhejiang Province (Q22C158467).

**Data Availability Statement:** The raw data supporting the conclusions of this article will be made available by the authors on request.

**Conflicts of Interest:** The authors declare that they have no conflicts of interest.

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
