# Peer review of "Effects of Progressive Drought Stress on the Growth, Ornamental Values, and Physiological Properties of Begonia semperflorens"

_horticulturae, doi:10.3390/horticulturae10040405_

Round 1

Reviewer 1 Report

Comments and Suggestions for Authors

There are quality standards for a finished ornamental plant of B. semperflorens

How many months is the useful life of B. semperflorens ornamental plants?

It is necessary to mention the substrate used, pH, porosity.

Every when they carried out the irrigation and the schedule

Why determine photosynthetic rates between 9:00 and 11:00 a.m.

In what environmental conditions did the plants remain, and after how many months or days did the test end?

What were the environmental conditions in which the plants remained (temperature, relative humidity, luminosity)

Coomasie brilliant blue dye, what structures it stains and how it is valued

Why use Duncan's multiple range statistical test and at that level of significance

Some references are incomplete

Citations appear complete, not by number, and arranged in alphabetical order.

Reviewer 2 Report

Comments and Suggestions for Authors

The manuscript aimed to describe the metabolic adaptation of Begonia semperflorens to progressive Drought Stress. The work is interesting, although the subject is not new

The authors carried out a lot of analyses, But they forgot to explain the methods they used.

In the protocols of drought stress, it is not described the frequency of irrigation, and other climatic parameters, to be more comprehensible obtained results. In particular, without having more detailed information about the experimental design, it is very strange that “ extremely light drought (ELD, 80% of normal water quantity)” produces statistical differences with control in almost parameters.

Abstract: specify the acronym used for the different treatments; The acronym with volume is not comprehensible

Material and methods:

The authors indicated: Line 84: Ten flowerpots of plants were used for each treatment

Line 104-105: Three biological duplications were performed in each experiment, and 3-5 flowerpots of plants were used 105 in each duplication

However, It is not clear the amount of biological replicated you used, and if the same samples were used for all the analyses.

Please better describe the choice and the number of plants used , and specifiy if the branches of the samples belonged to different plants

Rewrite the sentences, in the figure and tables, the number of replicates.

Line 76:specify cultivated under a consistent condition.

Line 77: specify the frequency of the irrigation with Hoagland solution and with water only

Line 92- 104: specify the amount of grams used for each sample; specify better the protocols used.

Results:

Line 122: water or solution?

The results contain a lot of values that clarify the figure, but the reading is not easy. On the other hand, the figures allow the readers to find the correlation of drought to change in metabolism.

However, no mention is made of the difference in the statistical analysis, which is very surprising: the samples in almost analyses are different from each other.

I am wondering about the So, to dispel some doubts, it is very important to have more samples and a figure of more plants treated with drought.

Figures: the legend needs an explanation of each acronym: :  in Figures1,6,7 are necessary.

Legend of table1: indicate the unit used for each morphological index

References: follow the indication of the journal

Round 2

Reviewer 1 Report

Comments and Suggestions for Authors

The suggestions were to be incorporated into the manuscript, and could be explained more broadly on a plant-environment biochemical and physiological basis.

The porosity of the substrate does not correspond to sand, according to the data it is very compact, it is not understandable how the roots could invade this substrate.

The light intensity was high, for being in greenhouse conditions. It is the average of how much time and hours logged outdoors or under cover.

The pH of the substrate, the data is from an alkaline substrate, the pH of the irrigation solution (Hoagland:water=1:2) should be pH 5.5.
